# Antibacterial Activity and Mechanism of Peptide PV-Q5 against *Vibrio parahaemolyticus* and *Escherichia coli*, Derived from Salt-Fermented *Penaeus vannamei*

**DOI:** 10.3390/foods12091804

**Published:** 2023-04-26

**Authors:** Jingyi Dai, Ritian Jin, Jialong Gao, Jude Juventus Aweya, Rong Lin, Guiling Li, Shen Yang

**Affiliations:** 1College of Ocean Food and Biological Engineering, Fujian Provincial Key Laboratory of Food Microbiology and Enzyme Engineering, Jimei University, 43 Yindou Road, Xiamen 361021, China; 2College of Food Science and Technology, Guangdong Provincial Key Laboratory of Aquatic Product Processing and Safety, Guangdong Ocean University, Zhanjiang 524088, China

**Keywords:** antimicrobial peptide, *Penaeus vannamei*, *Vibrio parahaemolyticus*, *Escherichia coli*, antibacterial mechanism

## Abstract

The increasing threat posed by antibiotic-resistant pathogens has prompted a shift to the use of naturally-derived antimicrobial peptides (AMPs) in place of chemical preservatives in controlling foodborne pathogens. In this study, ten peptides were identified from salt-fermented shrimps (*Penaeus vannamei*) using ultra-performance liquid chromatography-mass spectrometry. One of the peptides, designated PV-Q5 (QVRNFPRGSAASPSALASPR), with most features of an AMP, was further explored and found to possess strong antibacterial activity against *Vibrio parahaemolyticus* and *Escherichia coli*, with a minimum inhibitory concentration of 31.25 μg/mL. Moreover, PV-Q5 increased bacterial cell membrane permeability and ruptured bacteria cell membranes, as revealed by transmission electron microscopy. Circular dichroism analysis showed that the conformation of PV-Q5 was a random coil in phosphate-buffered saline and α-helical in sodium dodecyl sulfate, which is conducive for interaction with bacteria cell membranes. These findings indicated that PV-Q5 could find potential use in food preservation to control foodborne pathogenic bacteria.

## 1. Introduction

Generally, most foodborne diseases are caused by pathogenic microorganisms, which account for nearly 70% of global foodborne infections yearly [1]. Among these pathogenic foodborne microbes, bacteria are the most common pathogens associated with serious foodborne disease outbreaks and are responsible for the highest number of hospitalizations (63.9%) and deaths (63.7%) globally [2]. The halophilic Gram-negative bacterium *Vibrio parahaemolyticus* is an important foodborne pathogen that causes food poisoning and infections, characterized by abdominal pain, nausea, and vomiting [3]. *Escherichia coli* are bacteria species that are commonly found in animal and human gut. Although most *E. coli* strains are harmless, some pathogenic strains cause diseases in the gastrointestinal tract and respiratory pneumonia [4]. As a rich nutrient source, food provides a breeding ground for all types of *E. coli*, some of which have caused foodborne disease outbreaks globally [5]. Thus, these two foodborne pathogens are of public health concern in the food sector [6,7].

Chemical-based preservatives are mostly used in the control of foodborne pathogenic microorganisms [8]. Unfortunately, some chemical-based preservatives are perceived as “unhealthy”, such as the potential harm of long-term consumption to the human body and the emergence of some drug-resistant strains [9]. For this reason, alternative methods and agents that can preserve foods but limit the evolution of resistant strains are being explored. Some of the agents being explored include naturally produced bioactive substances, such as antimicrobial peptides (AMPs), that are produced as part of the innate immune response in many organisms and possess specific antibacterial activity, low resistance, and few toxic side effects [10,11]. In addition to the extraction of naturally-produced AMPs from organisms, salting has been a common traditional food processing and preservation method because it induces the production of peptides, some of which have antibacterial activity [12]. For instance, the peptide RHGYM was identified from Spanish dry-cured hams and found to possess strong antibacterial activity against *Listeria monocytogenes* (MIC of 4.14 mg/mL), indicating that AMPs could be potentially produced in cured meat products [13].

In the current study, we screened for peptides with antibacterial activity from salt-cured *Penaeus vannamei* using ultra-performance liquid chromatography-mass spectrometry (UPLC-MS) and in silico analysis. Among the low molecular weight peptides identified, the antibacterial activity of one of the peptides (designated PV-Q5) with antibacterial potential was explored against four pathogenic bacteria, Gram-negative (*V. parahaemolyticus* and *E. coli*) and Gram-positive (*S. aureus* and *S. iniae*).

## 2. Materials and Methods

### 2.1. Materials and Reagents

The bacteria strains used (i.e., *Vibrio parahaemolyticus* ATCC 17802, *Escherichia coli* ATCC 15224, *Staphylococcus aureus* ATCC 27217, and *Streptococcus iniae* ATCC 29178) were obtained from the China Center of Industrial Culture Collection (Beijing, China). All bacteria were cultured in nutrient broth (NB) medium at 37 °C. The human liver LO2 cell line was obtained from the Chinese Academy of Science Type Culture Collection (Shanghai, China) and cultured in Dulbecco’s modified eagle’s medium (DMEM) with 15% fetal bovine serum (FBS) at 37 °C in 5% CO_2_. DMEM and FBS were obtained from Gibco Co., Ltd. (Carlsbad, CA, USA). Shrimp (*Penaeus vannamei*) were purchased from a local supermarket (Xinhuadu Super Center, Xiamen, China). O-nitrophenyl-β-D-galactopyranose (ONPG), sodium dodecylbenzene sulfonate (SDS), sodium chloride (NaCl), M9 medium, and nutrient broth (NB) were purchased from LABLEAD Inc. (Beijing, China). Other analytical grade reagents (i.e., methanol, acetonitrile, lead citrate, uranyl acetate, and glutaraldehyde) were purchased from Xilong Technology Co., Ltd. (Shantou, China).

### 2.2. Screening and Identification of Peptides with Antimicrobial Peptides

*P. vannamei* were washed three times with cool distilled water and chopped into small pieces before being mixed with NaCl (30%, *w*/*w*), followed by fermentation in a laboratory fermenter (HB-EU 210, Holves Technology Co., Ltd., Beijing, China) at 37 °C for 15 days. Following this, the fermentation broth was centrifuged at 10,000 rpm/min for 10 min and desalted by solid phase extraction using a C18 cartridge (Sep-Pak, Waters, Milford, MA, USA), after which the desalted samples were digested overnight with mass-grade trypsin before being filtered through a 3 kDa ultrafiltration membrane. Next, samples were analyzed by ultra-performance liquid chromatography (UPLC, Nano Aquity UPLC system, Waters Corp, Milford, UT, USA) connected to a Thermo Q-Exactive mass spectrometer (Thermo Fisher Scientific, Milford, MA, USA) as previously described [14]. Briefly, 5.0 μL samples were injected onto a C18 column (Thermo Scientific Easy Column, 75 µm × 10 cm, 3 µm) used at a flow rate of 0.25 mL/min. Mobile phase A was 0.1% formic acid, while mobile phase B was acetonitrile. The program for the elution gradient was 0–1.0 min, 98% A, 1.0–55.0 min, 98% A–70% A, 55.0–60 min, 70% A–10% A. The mass spectrometry conditions were set as gas flow rate: 40 mL/min, auxiliary gas rate: 10 mL/min, spray voltage: 3.0 kV, capillary temperature: 300 °C, S-lens. 50%, HCD: 27%. Scan mode was positive ion, Full ms-ddms2, primary scan: resolution 70,000, range 350–1600 *m*/*z*, secondary scan: resolution 17,500, fixed first mass 120 *m*/*z*. Dynamic exclusion:10.0 s. The data obtained were searched against the *P. vannamei* protein database using the MAXQUANT (v1.6.5.0) software.

The identified peptides with antibacterial potential were predicted using in silico tools. The charge and hydrophobicity of peptides were calculated using the antimicrobial peptide calculator and predictor 3 (APD3) server (https://aps.unmc.edu/AP/prediction/prediction_main.php) (accessed on date 2 July 2020), while the antimicrobial characteristics of peptides were identified using the collection of anti-microbial peptides (CAMP) database (http://www.camp.bicnirrh.res.in/predict/) (accessed on date 5 August 2020). The 3D structure of peptides was predicted by ab initio calculations using the I-TASSER server (https://seq2fun.dcmb.med.umich.edu//I-TASSER/) (accessed on date 12 October 2021) [15].

### 2.3. Peptide Synthesis and Antibacterial Activity Determination

To determine the antibacterial activity of the peptides, PV-Q5 (QVRNFPRGSAASPSALASPR), we synthesized it by a commercial company (Scilight Biotechnology Co, Beijing, China) using the Fmoc solid-phase synthesis method as previously described [16]. The purity of the synthetic peptide determined by high-performance liquid chromatography (HPLC) was found to be 99%.

Different bacteria strains, i.e., *V. parahaemolyticus*, *E. coli*, *S. aureus*, and *S. iniae* were cultured in NB medium (8 g/L) at 37 °C for 24 h in a shaking incubator (New Brunswick, NY, USA). Bacteria cultures at the logarithmic growth phase, measured in terms of optical density at 600 nm (OD600), were used to determine the antibacterial activity of the peptides as previously described [17]. Briefly, 10^3^ CFU/mL of the bacteria were individually incubated with different concentrations of the peptides (i.e., 500–7.8 μg/mL) for 2 h at 37 °C. Next, 20 μL of the samples were spread onto NB agar plates before incubating at 37 °C for 24 h. Bacteria colonies were counted, and the minimum inhibitory concentration (MIC) was determined, defined as the peptide concentration that inhibited 80% of bacterial growth.

The bactericidal activity of PV-Q5 was analyzed using the time–kill curve, as previously described [18], with some modifications. Briefly, PV-Q5 was diluted to 1× MIC before being mixed with diluted *V. parahaemolyticus* and *E. coli* (10^3^ CFU/mL) and incubated at 37 °C. At different time points (i.e., 0, 1, 2, 3, 4, and 5 h), 20 μL of each sample was spread uniformly onto a glass culture dish and incubated at 37 °C for 24 h. The total number of bacteria on each plate was counted and recorded. Bacteria treated with sterile PBS (10 mM pH 7.4) were used as a control.

### 2.4. Bacteria Intracellular Membrane Permeability Determination

The ability of PV-Q5 to permeabilize the intracellular membrane of bacteria was determined using the level of o-nitrophenol produced as previously described [19]. Briefly, logarithmic growth phase bacteria (*V. parahaemolyticus* and *E. coli*) were collected by centrifugation (2700× *g* for 10 min), washed three times with PBS (10 mM pH 7.4) before being resuspended in 10 mL of sterile M9 lactose induction medium and incubated at 37 °C until the OD_600nm_ was greater than 0.4. Next, bacteria suspensions were mixed with PV-Q5 (final concentration at the MIC value) and 30 μL of 0.5 mg/mL ONPG before being incubated at 37 °C for 2 h with mixing followed by determining the OD_420nm_ at 1 h intervals for 8 h using a multimode plate reader (PerkinElmer, Shelton, CT, USA).

### 2.5. Transmission Electron Microscopy (TEM)

The effect of PV-Q5 on the ultrastructure of *V. parahaemolyticus* and *E. coli* was examined using TEM. First, logarithmic growth phase bacteria (10^3^ CFU/mL) were mixed with PV-Q5 (MIC value) and incubated at 37 °C for 2 h. Next, samples were collected by centrifugation (2700× *g* for 2 min) and washed three times with PBS (10 mM pH 7.4) before being fixed with 0.27 M glutaraldehyde at 37 °C for 12 h. Samples were then dehydrated with ethanol (30%, 50%, 70%, 80%, 90%, and 100%) before being treated with acetone for 20 min, followed by baking at 70 °C for 24 h. Thin slices (70–90 nm) were prepared on a copper grid and stained with lead citrate and uranyl acetate, after which samples were observed with an H-7650 transmission electron microscope (Hitachi, Tokyo, Japan).

### 2.6. Circular Dichroism (CD) Spectrophotometry

The secondary structure of PV-Q5 was determined using a Chirascan V100 circular dichroism chromatograph (Applied Photophysics Inc., London, UK) as previously described [20]. Briefly, PV-Q5 was dissolved in PBS (10 mM pH 7.4) or 25 mM SDS to a final concentration of 50 μg/mL. Next, CD spectra of samples were collected from 190 to 280 nm using a bandwidth of 1 nm and scanning speed of 100 nm/min at 25 °C.

### 2.7. Cytotoxicity of PV-Q5

The cytotoxicity of PV-Q5 on human cells was determined by the (4, 5-dimethylthiazol-2-yl)-2, 5-diphenyl-2H-tetrazolium bromide (MTT) method. Briefly, normal human hepatocytes (LO2 cells) were seeded into 96-well plates (5000 cells/well) and incubated at 37 °C for 24 h. After the media was aspirated, 100 μL of PV-Q5 (at different concentrations of 1×–16× MIC) was added to each well before being incubated at 37 °C for 24 h. Next, 20 μL of MTT (5 mg/mL) was added to each well and incubated at 37 °C for 4 h, after which the media was aspirated and 150 μL of dimethyl sulfoxide was added to each well. The absorbance at 490 nm was read on a Synergy H1 multimode plate reader (Biotech Instruments Inc., Winooski, VT, USA). Samples treated with PBS were used as control, while wells containing only the media and MTT reagent were used as blank. Cell viability was calculated using the following equation:$$\mathrm{Cell}\;\mathrm{viability}\;\left(\%\right)=\frac{OD_{test}-OD_{blank}}{OD_{control}-OD_{blank}}\times 100\%.$$

### 2.8. Statistical Analysis

All experiments were performed in triplicate, and data were presented as mean ± standard deviation. One-way ANOVA was performed using the SPSS software (version 22.0) and GraphPad Prism software (version 8.3). Differences were considered statistically significant at *p* < 0.05.

## 3. Results and Discussion

### 3.1. Screening and Prediction of Peptides with Antibacterial Activity

Ten peptides of molecular weights (MW) < 3 kDa were identified from salt-fermented *P. vannamei* (Table 1). In silico analysis revealed that identified peptides had an average of 14.5 amino acids, molecular weights of 973 to 2116 Da, hydrophobicity of 20 to 44%, and a net charge of −3 to +3. All the sequences of the ten peptides were present in the proteomes of *P. vannamei.*

During food fermentation, bioactive peptides are accumulated due to the activities of endogenous hydrolases and the metabolic activity of salt-tolerant or halophilic microorganisms that contribute to protein hydrolysis [21]. Although the efficiency of peptide production through natural salt fermentation is lower than that of hydrolysis by proteases, it indicates that salt fermentation could generate AMPs [22]. Generally, peptides with net charges of +2 to +9 and hydrophobicity between 30 and 60% have excellent antimicrobial properties [23]. In addition, most short-chain Pro-rich peptides exhibit potent activity against Gram-negative bacteria, while Gly-rich peptides have increased selectivity while retaining high antimicrobial activity [24]. In the current study, one of the identified, designated PV-Q5 (QVRNFPRGSAASPSALASPR) had a +3 net charge, 35% hydrophobicity, 20% glycine and proline ratio, and 0.800 antimicrobial peptide prediction score. This indicates that PV-Q5 possesses most of the characteristic properties of AMPs; hence, it was chosen for further evaluation.

### 3.2. Antibacterial Activity of PV-Q5

To explore the antibacterial activity of PV-Q5 against a selection of pathogenic Gram-negative (*V. parahaemolyticus* and *E. coli*) and Gram-positive (*S. aureus* and *S. iniae*) bacteria, the MIC was determined. As shown in Table 2, PV-Q5 had MICs > 500 μg/mL against *S. aureus* and *S. iniae*, while that against *V. parahaemolyticus* and *E. coli* was 31.25 μg/mL. These results indicate that PV-Q5 is more efficient against having a strong antibacterial effect on Gram-negative bacteria (*V. parahaemolyticus* and *E. coli*) compared with Gram-positive bacteria (*S. aureus* and *S. iniae*). Given the highest MIC of PV-Q5 against *V. parahaemolyticus* and *E. coli*, we further analyzed the antibacterial activity using a time–kill curve. The results from the time–kill kinetic analysis revealed that when *V. parahaemolyticus* was treated at the MIC of PV-Q5, the bacterial numbers reduced by 64.83% within 2 h and up to 87.36% in 5 h, compared with the control (Figure 1A). Similarly, 1× MIC of PV-Q5 was able to reduce the number of *E. coli* by 70.53% in 2 h and by 87.42% in 5 h, compared with the control (Figure 1B). These results further show that PV-Q5 has a strong antibacterial effect on *V. parahaemolyticus* and *E. coli*.

Not all AMPs have broad-spectrum activity against Gram-negative and Gram-positive bacteria. Thus, the observed selective antibacterial activity of PV-Q5 against *V. parahaemolyticus* and *E. coli* (Gram-negative bacteria) is consistent with other AMPs. For example, Nisin, the naturally-derived cationic AMP produced by *S. lactis* and widely used as a food preservative, only has strong activity against Gram-positive bacteria (e.g., *S. aureus* and *S. pneumoniae*) and not against Gram-negative bacteria [25]. Similarly, rBPI 21, an AMP derived from the human neutrophil BPI protein, displays strong bactericidal activity against Gram-negative bacteria (e.g., *E. coli*) but weak activity against Gram-positive bacteria (e.g., *S. aureus*) [26]. Interestingly, PV-Q5 and one of our previously characterized peptides, PvHS9 [27], also isolated from the same shrimp species (*P. vannamei*), have different MIC against *V. parahaemolyticus*, i.e., the MIC of PV-Q5 (31.25 μg/mL) is lower than PvHS9 (62.5 μg/mL). The antibacterial activity of PV-Q5 and its mechanism of action against *V. parahaemolyticus* warrants further investigation.

### 3.3. Effect of PV-Q5 on Bacteria Membrane Permeability

The ability of PV-Q5 to permeabilize bacteria membranes was examined in terms of the OD_420nm_ of O-nitrophenyl produced from O-nitrophenyl β-D-galactopyranoside after hydrolysis by bacteria membrane β-galactosidase [28]. When *V. parahaemolyticus* was treated with 1× MIC of PV-Q5, the OD_420nm_ was 0.68, while that of the control was 0.57 at 1 h (Figure 2A). Similarly, treatment of *E. coli* with 1× MIC for 1 h produced an OD_420nm_ of 0.65 compared to 0.53 of control (Figure 2B). Moreover, the level of O-nitrophenyl produced in terms of OD_420nm_ upon treatment of both *V. parahaemolyticus* and *E. coli* was higher at 2× MIC of PV-Q5 compared with ½ × MIC, indicating that bacteria membrane permeability increases concentration dependently (Figure 2A,B). Thus, PV-Q5 can damage the intracellular membrane of *V. parahaemolyticus* and *E. coli*.

Some AMPs can replace Mg^2+^ in the outer lipopolysaccharide membrane of Gram-negative bacteria to destabilize bacteria’s outer surface, which enhances the penetration of AMPs to disrupt the intracellular membrane [29]. Cationic AMPs can interfere with the formation and accumulation of negatively charged extracellular polymers between bacteria cells to change their inner membrane permeability, hence resulting in bacteria death [30].

### 3.4. Effect of PV-Q5 on Bacteria Membrane Ultrastructure

Transmission electron microscopy (TEM) was used to examine the effect of PV-Q5 on the membrane ultrastructure of *V. parahaemolyticus* and *E. coli*. When bacteria were treated with 1× MIC of PV-Q5 or PBS for 2 h followed by TEM examination, the ultrastructures of the cell membranes of *V. parahaemolyticus* and *E. coli* were smooth and fully dense in the PBS control group (Figure 3A,C). On the other hand, treatment with PV-Q5 caused severe destruction and deformation of *V. parahaemolyticus*, changing their spherical shape, into irregular bulged shapes, with the middle cell layer becoming loose compared with the control (Figure 3B). Similar structural destruction was observed in *E. coli* cells treated with PV-Q5, where the cytoplasmic cell density was reduced, and the cell membranes were damaged and deformed (Figure 3D). Damage to the bacterial cell membranes is manifested by the loss of lipid layer structure and a decrease in electron density [31], which was a similar phenomenon observed when *V. parahaemolyticus* and *E. coli* were treated with PV-Q5. The destruction of bacteria ultrastructure by PV-Q5 could be due to changes in bacterial membrane permeability, leading to their disruption and breakdown [32].

### 3.5. Changes in the Secondary Structure of PV-Q5

To observe the changes in the secondary structure of PV-Q5 under different conditions, we dissolved it in PBS (10 mM, pH 7.4) or SDS (25 mM) and examined it by circular dichroism (CD) spectroscopy. As shown in Figure 4A, the CD spectrum of PV-Q5 in PBS shows a negative absorption peak near the 200 nm wavelength with a random coil secondary structure. On the other hand, when PV-Q5 was in an SDS solution, it exhibited a positive absorption peak near 185 nm and negative absorption peaks near 200 nm and 220 nm, which are features of an α-helical structure (Figure 4A). These results indicated that the secondary structure of PV-Q5 undergoes transformation under different fluid conditions.

To facilitate the interaction between AMPs and bacterial cell membranes, the secondary structure of AMPs undergoes a rapid transformation when in contact with membranes [33]. The secondary structure of most AMPs exhibits random coil in PBS solution, but when in SDS solution, the hydrogen or hydrophobic bonds of AMPs unfold, changing the interaction between nonpolar side chains to form aggregates, which affects the formation of hydrogen bonds between polar side chains, and therefore induces helical structure formation [34]. In PV-Q5, the SALAS sequence contains serine (S) in the head and tail, which is an amino acid with a polar group, while the alanine (A) and leucine (L) in the middle are amino acids with nonpolar hydrophobic side chains. Thus, SDS can cause the serine-containing polar group to form a hydrogen bond, whereas the nonpolar hydrophobic side chains of alanine and leucine can aggregate with each other to curl and fold into an α-helical structure [35]. For this reason, PV-Q5 can adapt structurally from a random coil to α-helical when it binds to the liposomes of bacterial cell membranes to exert its antibacterial effect. Three-dimensional structural modeling predictions confirmed the topology for PV-Q5 (Figure 4B). PV-Q5 was predicted to have an α-helix motif, the site where it curls and folds into an α-helix is at the exact location of the five amino acids in the sequence (i.e., SALAS), which is essential for antibacterial activity.

Most of the reported AMPs are α-helical with hydrophilic and hydrophobic side chains on both sides of the α-helix that promotes interaction with bacteria cell membranes [36]. When α-helical AMPs bind to bacteria membranes, they aggregate to form pores on the membrane surface, which results in the outflow of cellular contents, a decrease in cytoplasmic density, and cell death [37]. PV-Q5 also contains three arginine residues (positively charged) and several other hydrophobic amino acid residues, which could also contribute to its antibacterial activity. The positively charged arginine residues promote electrostatic interactions, whereas the hydrophobic amino acid residues (i.e., phenylalanine, alanine, and proline) enhance hydrophobic interactions between PV-Q5 and bacterial membranes to facilitate the entry of PV-Q5 into the phospholipid bilayer of bacteria, to depolarize and disrupt the membranes and cause cell death.

### 3.6. Cytotoxicity Analysis of PV-Q5

To explore the cytotoxic effect of PV-Q5 on human cells, we examined the cytotoxicity of PV-Q5 on normal human cells (LO2 cells) using the MTT assay [38]. When LO2 cells were treated with various concentrations of PV-Q5 (MIC to 8× MIC), the cell viability was 95.3 ± 0.82% to 89.3 ± 1.7% (Figure 5). Even at a very high concentration of PV-Q5 (i.e., 16× MIC), the cell viability was still 88.3%, indicating that PV-Q5 has very low toxicity on LO2 cells. Moreover, the effect of PV-Q5 on cell viability was not significantly different from that of Nisin (Figure 5).

Nisin is one of the AMP used in food preservation that has GRAS (generally considered as safe) status from both WHO and FDA [39]. Thus, given the strong antibacterial activity of PV-Q5 and its low toxicity on human cells (LO2 cells), it indicates that PV-Q5 could potentially be an important natural AMP (for Gram-negative bacteria) with the possibility of GRAS status since many bacteria have become more resistant to traditional antibiotics [40].

## 4. Conclusions

In this study, ten low molecular weight peptides (less than 3 kDa) were identified in salt-fermented shrimp (*P. vannamei*), among which one peptide designated PV-Q5 with most features of an AMP was further explored. PV-Q5 had strong antibacterial activity against *V. parahaemolyticus* and *E. coli*, both with MIC of 31.25 μg/mL. Structurally, PV-Q5 exhibited random coil in PBS and α-helical in SDS solution to facilitate its interaction with bacterial cell membranes. PV-Q5 could permeabilize bacteria membranes, destroying them and decreasing cytoplasmic density, hence causing bacteria death. These findings provided new insights into the potential screening for naturally-derived AMPs from salt-fermented food products for application in food preservation. Future studies would further explore the use of PV-Q5 in preserving some specific foods and also the reason for the selective action of PV-Q5 on Gram-negative bacteria and not Gram-positive bacteria.

## Figures and Tables

**Figure 1 foods-12-01804-f001:**
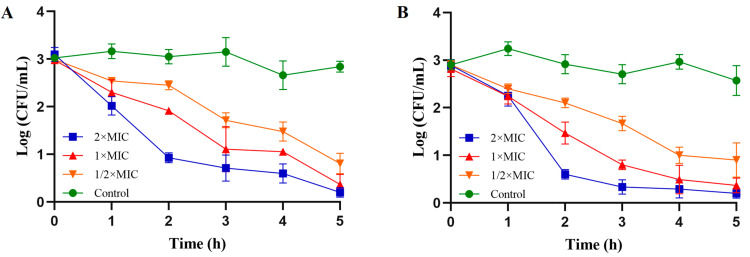
Time–kill kinetics of PV-Q5 against *V. parahaemolyticus* and *E. coli.* Samples were treated with PBS (control) or different concentrations (1× MIC, 2× MIC, and ½× MIC) of PV-Q5 and (**A**) *V. parahaemolyticus* or (**B**) *E. coli.* The MIC used was 31.25 μg/mL.

**Figure 2 foods-12-01804-f002:**
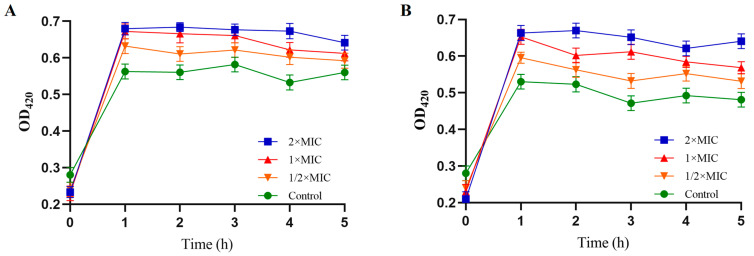
Inner membrane permeability of *V. parahaemolyticus* and *E. coli.* After treatment with PBS (control) or different concentrations (1× MIC, 2× MIC, and ½× MIC) of PV-Q5 and (**A**) *V. parahaemolyticus* or (**B**) *E.coli.* The MIC of PV-Q5 used was 31.25 μg/mL.

**Figure 3 foods-12-01804-f003:**
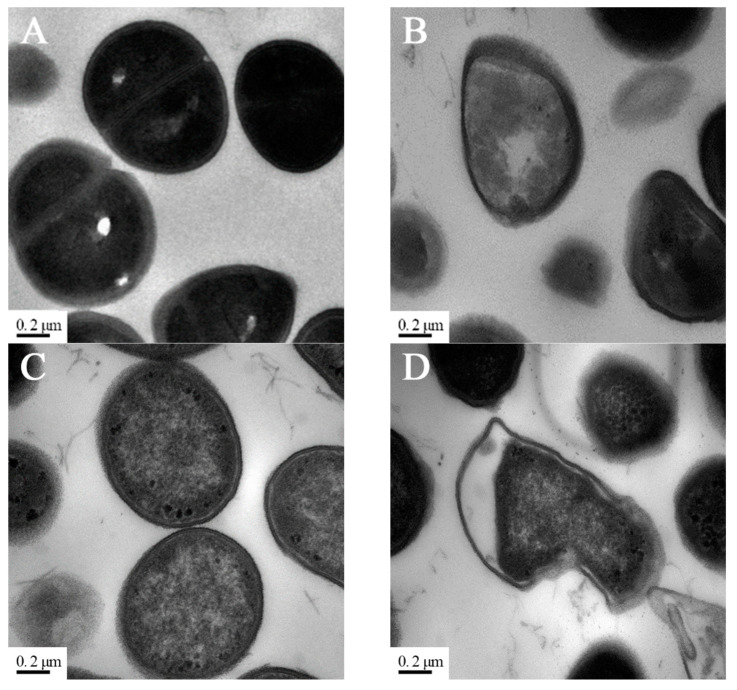
TEM images of bacteria after treatment with PV-Q5. *V. parahaemolyticus* (10^3^ CFU/mL) treated with (**A**) PBS (control) or (**B**) 2× MIC of PV-Q5 for 2 h. *E. coli* (10^3^ CFU/mL) treated with (**C**) PBS (control) or (**D**) 2× MIC of PV-Q5 for 2 h. The MIC used was 31.25 μg/mL.

**Figure 4 foods-12-01804-f004:**
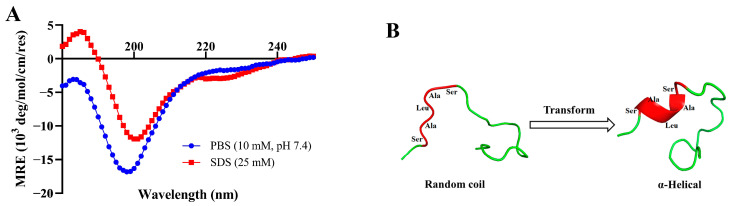
Structural changes of PV-Q5 under different conditions. Changes in the structure of PV-Q5 after treatment with (**A**) PBS (10 mM pH 7.4) or SDS (25 mM). (**B**) Predicted changes in the tertiary structure of PV-Q5.

**Figure 5 foods-12-01804-f005:**
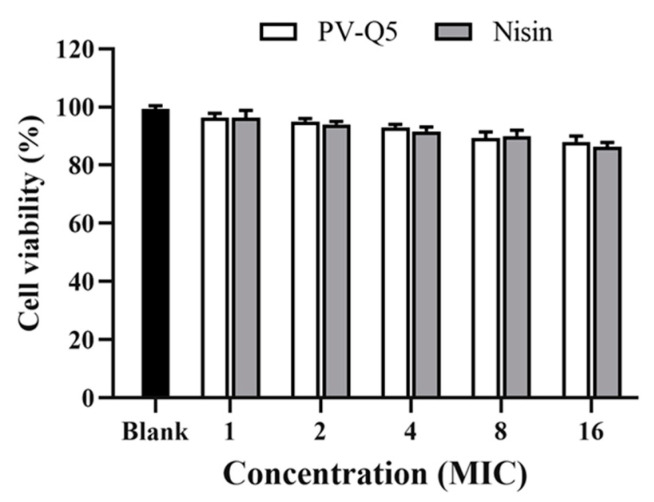
Effect of PV-Q5 on the viability of normal human cells (LO2 cells). The effect of different concentrations (1× MIC–16× MIC) of PV-Q5 or Nisin on cell viability was determined using the MTT assay. The MIC of PV-Q5 was 31.25 μg/mL. Nisin was used as positive control and PBS as negative control.

**Table 1 foods-12-01804-t001:** Predicted antibacterial potential of peptides isolated from salt-fermented *P. vannamei*.

No.	Peptide Sequences	MW (Da)	Hydrophobic Ratio (%)	Net Charge	AMP Probability
PV-Q1	TREQLAEEKK	1105.7	20	0	0.015
PV-Q2	HADAQKNLR	1051.6	33	+2	0.079
PV-Q3	KAISSLEAR	973.6	44	+1	0.011
PV-Q4	KDAYVGDEAQSK	1170.6	25	−1	0.065
PV-Q5	QVRNFPRGSAASPSALASPR	2068.1	35	+3	0.800
PV-Q6	TGSNVFDMFTQKQVAEFK	2076.0	38	0	0.163
PV-Q7	RQIEEAEEIAALNLAKYR	2116.1	44	−1	0.369
PV-Q8	REVFELGGVTHLLNVLK	1923.1	47	+1	0.289
PV-Q9	AGFAGDDAPRAVFPSIVGR	1902.0	47	0	0.347
PV-Q10	IQEKEEEFDATR	1493.7	25	−3	0.341

**Table 2 foods-12-01804-t002:** Antibacterial activity of peptide PV-Q5 against Gram-negative and Gram-positive bacteria.

Peptides	MIC (μg/mL)			
	Gram-Negative		Gram-Positive	
	*V. parahaemolyticus*	*E. coli*	*S. aureus*	*S. iniae*
QVRNFPRGSAASPSALASPR	31.25	31.25	>500	>500

## Data Availability

No new data were created or analyzed in this study. Data sharing is not applicable to this article.

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
