# Peer review of "Antibacterial Activity and Mechanism of Peptide PV-Q5 against Vibrio parahaemolyticus and Escherichia coli, Derived from Salt-Fermented Penaeus vannamei"

_foods, 2023, doi:10.3390/foods12091804_

Round 1

Reviewer 1 Report

In this study, the authors identified ten peptides from shrimp fermented with salt (P. vannamei), the peptide called PV-Q5 had strong antibacterial activity against the Gram negative, in particular V. parahaemolyticus and E. coli,  resulting in a potential natural product for food preservation. The paper is  interesting, especially  for the possible application in food preservation but I have some doubts about some experiments.

1a)It is not clear in the determination of the mic the method used (plate Colony count method). You can add reference and explain it better this section?

1b)Why did you use NB?

2a)In the time killing experiment the authors show a reduction in bacterial load over time. It is strange that at the value of 1/2 mic there is after 5 hours a complete reduction of the bacterial charge almost like the mic value. Are the authors sure of this result?

2b)The authors report negative control (pbs and bacterium) in anti-bacterial experiments. Didn’t you also use a standard positive control, like an antibiotic?

I suggest you this recent paper for mic and time killing experiments.(https://doi.org/10.3390/pharmaceutics15020700; https://doi.org/10.1128/Spectrum.00152-21;  https://doi.org/10.1016/j.micres.2022.126980)

Author Response

In this study, the authors identified ten peptides from shrimp fermented with salt (P. vannamei), the peptide called PV-Q5 had strong antibacterial activity against the Gram negative, in particular V. parahaemolyticus and E. coli, resulting in a potential natural product for food preservation. The paper is interesting, especially for the possible application in food preservation but I have some doubts about some experiments.

Comment 1. It is not clear in the determination of the mic the method used (plate Colony count method). You can add reference and explain it better this section?

I suggest you this recent paper for mic and time killing experiments.

https://doi.org/10.3390/pharmaceutics15020700; https://doi.org/10.1128/Spectrum.00152-21;  https://doi.org/10.1016/j.micres.2022.126980

Response 1: Thanks for your valuable comments, suggestions, and references provided. We have revised this section of the materials and method 2.3, which now reads as follows:

Different bacteria strains, i.e., V. parahaemolyticus, E. coli, S. aureus, and S. iniae were cultured in NB medium (8 g/L) at 37 °C with shaking for 24 h in an incubator (New Brunswick, New York, USA). Bacterial cultures at the logarithmic growth phase, measured in terms of optical density at 600 nm (OD600 nm), were used to determine the antibacterial activity of the peptide as previously described (Xedzro et al., 2022. Microbiological Research, 258, 126980). Briefly, 103 CFU/mL of the bacteria were incubated with different concentrations of the peptides (i.e., 500 - 7.8 μg/mL) for 2 h at 37°C. Next, 20 μL of the samples were spread on NB agar plates before being incubated at 37°C for 24 h. Bacteria colonies were counted, and the minimum inhibitory concentration (MIC) determined, defined as the peptide concentration inhibiting 80% of bacterial growth.

Comment 2. Why did you use NB?

Response 2: We appreciate the reviewer’s comment. NB culture medium, composed of 5% NaCl at a pH of 7.2 ± 0.2, nitrogen, vitamins, amino acids, and carbon sources with peptone and beef extract, is a common medium used to grow bacteria (Wang M. et al., 2020. Journal of Materials Science & Technology, 70, 224-232). For instance, halophilic bacteria such as V. parahaemolyticus grow in media of pH of 7.0-9.5 and 3.3-37% NaCl (Ndraha, N. et al., 2022. Current Opinion in Food Science, 100927). Similarly, E. coli grows rapidly in NB media (Wahyuningsih, N., & Zulaika, E., 2019. Jurnal sains dan Seni ITS, 7(2), 36-38).

Comment 3. In the time killing experiment the authors show a reduction in bacterial load over time. It is strange that at the value of 1/2 MIC there is after 5 hours a complete reduction of the bacterial charge almost like the mic value. Are the authors sure of this result?

Response 3: Thanks for raising this issue via this comment. The MIC values are accurate, as evidenced by the 80% reduction in bacterial numbers after 2 h of treatment with the peptides at the MIC concentration (31.25 μg/mL). As to why the after 5 h the bacterial reduction upon treatment with ½ MIC and MIC are similar, the reason could be that some antimicrobial peptides at lower concentrations exert their effects slowly and therefore a similar antibacterial effect after prolonged treatment. We hope our explanation helps clarify the point being raised by the reviewer.

Comment 4. The authors report negative control (pbs and bacterium) in anti-bacterial experiments. Didn’t you also use a standard positive control, like an antibiotic?

Response 4: We appreciate the reviewer’s comment. Generally, Nisin is used as a positive control for research on foodborne Gram-positive bacteria, but since the peptide explored in our study has antibacterial effects on Gram-negative bacteria, we did not use niasin or other antibiotics. The absence of a positive control does not affect the data and its interpretation since many previous studies have also only used negative controls, such as PBS (Luo, X. et al., 2022. Food Bioscience, 49, 101903).

Reviewer 2 Report

The new peptide PV-Q5 with antibacterial properties was purified and studied. Low toxicity and activity against gram- strains were demonstrated. The possible mechanism of PV-Q5 action was also investigated and presented.

Several experiments should be done:

1) Resistant clone selection - it will help to estimate the frequency of resistance occurrence and following sequencing of the resistant clones will help to understand more about the mechanism of action.

2) Mutagenesis of PV-Q5 - substitution of the most important amino acids to alanine and testing of the activity of the mutant peptides.

Author Response

Reviewer 2

The new peptide PV-Q5 with antibacterial properties was purified and studied. Low toxicity and activity against gram- strains were demonstrated. The possible mechanism of PV-Q5 action was also investigated and presented.

Several experiments should be done:

Comment 1. Resistant clone selection - it will help to estimate the frequency of resistance occurrence and following sequencing of the resistant clones will help to understand more about the mechanism of action.

Response 1: We appreciate the reviewer’s comments and the point on resistant clone selection. The main aim of this study is to show that potent antimicrobial peptides can be obtained from salt-fermented shrimp, which is a fast way of obtaining foodgrade peptides for application in the food industry. Our main aim is not whether the bacteria used for the study will develop resistance to these peptides. Nonetheless, we know that the issue of antimicrobial resistance is of global concern, but it has also been shown that bacteria do not easily develop resistance to antimicrobial peptides like antibiotics (Andersson, D. I., 2016, Drug Resistance Updates, 26, 43-57; Armas, F., 2021, Int. J. Mol. Sci., 22, 7959; Galdiero, E., 2019, Pharmaceutics, 11, 322; Wu, Q., 2018, Toxins, 10, 461; Mahlapuu, M., 2020, Critical Reviews in Biotechnology, 40(7), 978-992). In any case, we will consider this advice from the reviewer in our future application studies.

Comment 2. Mutagenesis of PV-Q5 - substitution of the most important amino acids to alanine and testing of the activity of the mutant peptides.

Response 2: Thanks for raising this point via this comment. In most of our previous in silico analyses and that of other researchers, the use of scanning or mutations have been found to affect or change the antimicrobial activity because replacing most of the important amino acids with alanine will change the hydrophobicity and net charge of the peptides (Migoń, D. et al., 2019. Probiotics and antimicrobial proteins, 11, 1042-1054), which are key factors that determine the activity of antimicrobial peptides. Thus, we think that adding or performing this experiment would not add or change the key findings of this study. We hope our response helps clarify this point.

Reviewer 3 Report

The paper presents a description of the natural AMP derived from salt fermented Penaeus vannamei.  Nowadays, the AMP can be considered an important alternative to classic antimicrobial substances and the this study proposed a new  peptide  active against Vibrio parahaemplyticus and E.coli laying the basis for a future study to test a new solution to extend the preservation of food. Despite I propose some revision:

in the introduction  the authors highlighted the role of Vibrio parahaemolyticus and E.coli as causes of foodborne disease, but the description of  E Coli is focalised on  Shiga toxin-producing E.coli, that were not the object of study.

Materials e methods.

2.3: Can the Authors better detail the preparation of microbiological tests against Vibrio parahaemolyticus and E.coli

What about Gram positive bacteria?

Finally, I suggest broadening the conclusions by a more complete description of future developments and applications of AMP

Author Response

Reviewer 3

The paper presents a description of the natural AMP derived from salt fermented Penaeus vannamei. Nowadays, the AMP can be considered an important alternative to classic antimicrobial substances and this study proposed a new peptide active against Vibrio parahaemplyticus and E. coli laying the basis for a future study to test a new solution to extend the preservation of food. Despite I propose some revision:

Comment 1. In the introduction the authors highlighted the role of Vibrio parahaemolyticus and E. coli as causes of foodborne disease, but the description of E. coli is focalised on Shiga toxin-producing E. coli, that were not the object of study.

Response 1: Thanks for the valuable comment. We have revised the introduction, especially the description of E. coli. The revised sentence now reads as follows:

Escherichia coli are bacteria species that are commonly found in animal and human gut. Although most E. coli strains are harmless, some pathogenic strains cause diseases in the gastrointestinal tract and respiratory pneumonia [Ma, Y et al., 2019, Food Control, 106, 106712]. As a rich nutrient source, food provides a breeding ground for all types of E. coli, some of which have caused foodborne disease outbreaks globally [Aijuka, M. et al., 2019, Food Microbiology, 82, 363-370].

Comment 2. Materials methods. 2.3: Can the Authors better detail the preparation of microbiological tests against Vibrio parahaemolyticus and E. coli

Response 2: We appreciate the reviewer’s suggestion. In response, we have added the bacterial culture conditions to section 2.3 of the revised manuscript as follows:

Different bacteria strains, i.e., V. parahaemolyticus, E. coli, S. aureus, and S. iniae were cultured in NB medium (8 g/L) at 37 °C for 24 h in a shaking incubator (New Brunswick, New York, USA). Bacteria cultures at the logarithmic growth phase, measured in terms of optical density at 600 nm (OD600), were used to determine the antibacterial activity of the peptides as previously described [Xedzro, C. et al., 2022, Microbiol. Res. 258, 126980]. Briefly, 103 CFU/mL of the bacteria were individually incubated with different concentrations of the peptides (i.e., 500 - 7.8 μg/mL) for 2 h at 37 °C. Next, 20 μL of the samples were spread onto NB agar plates before incubating at 37 °C for 24 h. Bacteria colonies were counted, and the minimum inhibitory concentration (MIC) was determined, defined as the peptide concentration that inhibited 80% of bacterial growth.

Comment 3. What about Gram positive bacteria?

Response 3: Thanks for the comment. Since the peptide had an inhibitory effect mainly on Gram-negative bacteria we did not include Gram-positive bacteria in the mechanistic studies.

Comment 4. Finally, I suggest broadening the conclusions by a more complete description of future developments and applications of AMP

Response 4: We appreciate the reviewer’s comment. We have broadened the conclusion to include future applications of the peptide.

Reviewer 4 Report

I reviewed an Article entitled Antibacterial activity and mechanism of peptide PV-Q5 against Vibrio parahaemolyticus and Escherichia coli, derived from salt- fermented Penaeus vannamei and find to be more suitable for some antimicrobial journal.

Author Response

Reviewer 4

Comment 1. I reviewed an Article entitled Antibacterial activity and mechanism of peptide PV-Q5 against Vibrio parahaemolyticus and Escherichia coli, derived from salt-fermented Penaeus vannamei and find to be more suitable for some antimicrobial journal.

Response: Thanks for reviewing our paper. Although the paper explored the antibacterial activity and action mechanism of a peptide, we wish to draw the attention of the reviewer that food science or food microbiology covers various aspects along the food chain, which include food processing, preservation, and safety. Currently, most food preservation methods that are nonthermal use chemicals or antibiotics, which pose risk to both humans and the environment, given that antibiotic resistance has become a global emergency (Courvalin, P. 2016, Clinical Microbiology And Infection, 22, 405-407). for these reasons, the content of this paper falls within the aims and scope of this journal, i.e., Foods. We hope our explanation helps clarify the point raised by the reviewer.

Round 2

Reviewer 2 Report

Despite the authors' objections, resistant clone selection should be performed. Genome sequencing could be performed further - but at least the ratio of the resistant clones should be estimated. Authors should try to take more 10^10 cells and grow at a high concentration of compound (>5-10MIC) then calculate CFU.

Author Response

We appreciate the reviewer’s comment on performing resistant clone selection using >5-10xMIC. Although we did mention in our previous response that this is not the main aim of our study, we wish to indicate that when we treated bacteria with 2 x MIC or higher MICs, peptide PVQ-5 cleared all the bacteria on the plates (please see Figure 1 and 2 below). Thus, even if we wish to grow the bacteria in higher peptide concentration, as suggested by the reviewer, no bacteria will survive this high concentration to be selected for resistant clone screening. We hope the data provided below and our explanation clarifies the point being raised and that at MICs greater than 2 x MIC, no bacteria colonies are left for resistant clone selection and, for that matter, genome sequencing.

Figure 1: Effect of peptide PVQ-5 on V. parahaemolyticus

Figure 2: Effect of peptide PVQ-5 on E. coli

Reviewer 3 Report

I thanks the Authors to review the paper as I suggested. I propose to accept the paper  in present form.

Author Response

We appreciate reviewer’s comment. This would encourage us to further explore the bactericidal effect of AMPs against foodborne pathogenic bacteria in the future.